# What Drives the Influence of Health Science Communication Accounts on TikTok? A Fuzzy-Set Qualitative Comparative Analysis

**DOI:** 10.3390/ijerph192113815

**Published:** 2022-10-24

**Authors:** Ran Liu, Tianan Yang, Wenhao Deng, Xiaoyan Liu, Jianwei Deng

**Affiliations:** 1School of Public Health and Management, Wenzhou Medical University, Wenzhou 325035, China; 2School of Management and Economics, Beijing Institute of Technology, Beijing 100081, China; 3Sustainable Development Research Institute for Economy and Society of Beijing, Beijing 100081, China; 4School of Languages and Communication Studies, Beijing Jiaotong University, Beijing 100044, China

**Keywords:** TikTok, microvideo, health science communication, account influence, fsQCA

## Abstract

Medical institutions face a variety of challenges as they seek to enhance their reputation and increase the influence of their social media accounts. Becoming a social media influencer in the health field in today’s complex online environment requires integrated social and technical systems. However, rather than holistically investigating the mechanism of account influence, studies have focused on a narrow subset of social and technical conditions that drive online influence. We attribute this to the mismatch between complex causality problems and traditional symmetric regression methods. In this study, we adopted an asymmetric configurational perspective that allowed us to test a causally complex model of the conditions that create strong and not-strong account influence. We used fuzzy-set qualitative comparative analysis (fsQCA) to detect the effects of varying configurations of three social system characteristics (i.e., an oncology-related attribute, a public attribute, and comment interaction) and two technical system characteristics (i.e., telepresence and video collection) on the TikTok accounts of 63 elderly Chinese doctors (60 to 92 years old). Our results revealed two pathways associated with distinct sociotechnical configurations to strong account influence and three pathways associated with distinct sociotechnical configurations to not-strong account influence. Furthermore, the results confirmed that a single antecedent condition cannot, on its own, produce an outcome, i.e., account influence. Multiple inter-related conditions are required to produce an influential account. These results offer a more holistic picture of how health science communication accounts operate and reconcile the scattered results in the literature. We also demonstrate how configurational theory and methods can be used to analyze the complexities of social media platforms.

## 1. Introduction

A growing number of health institutions around the world publish videos about physical activities, healthy nutrition, stress relief, oral hygiene, vaccination, and injury prevention on their social media accounts. Their aim is to increase primary prevention and the health literacy of the general public [1,2]. In China, health communication is evolving via Internet videos; for example the Healthy China Strategy and 5G technologies provide policy guidance and technical platforms for the development of health science videos [3]. As of June 2021, there were 1.011 billion Internet users in China, comprising the largest digital society in the world [4]. According to iiMedia Research, the number of microvideo users in China reached 808 million in 2021 [5]. Founded in 2017, TikTok is one of the fastest-growing microvideo social media platforms, with professional video sharers from a variety of fields producing content via the vertical development style of the TikTok community. The total number of downloads of the Chinese and international versions of TikTok in the global app store has exceeded three billion in 2021, reigning as the global download champion [6]. TikTok has more than 600 million daily active users, with more than 400 million daily video searches [7]. Unsurprisingly, TikTok has become one of most representative public health communication channels, providing users with micro health-related videos [8]. In China, some of the most popular health accounts on TikTok are run by well-known elderly doctors, including Professor Huang Hanyuan, an 89-year-old from the Department of Breast Surgery of Peking Union Medical College Hospital; Professor Tian Lianzhen, an 80-year-old from the Department of Gynecology and Obstetrics of the Second Affiliated Hospital of Xi’an Jiaotong University; and Zhang Daizhao, a 92-year-old from the Department of Traditional Chinese Medicine Oncology of China-Japan Friendship Hospital. An increasing number of public health departments, medical institutions, and health professionals have begun to release health-related popular science videos on TikTok [8,9,10]. Although health science knowledge is also disseminated via traditional social media, such as Weibo and WeChat, the user base of microvideo social media, represented by TikTok, is gradually increasing, with middle-aged and elderly users increasingly representing main user groups [11]. Together with its advantages of easy editing and production of video content, social presence, and immersion, TikTok is gradually becoming a major platform for social interaction and dissemination of health knowledge by healthcare professionals [12,13]. It has played an important role in health communication and education during the COVID-19 pandemic [14]. Therefore, in this study, we focus on TikTok rather than other traditional social network platforms.

At the individual level, the use of TikTok by healthcare providers, as represented by elderly doctors, to disseminate health knowledge helps raise disease awareness among viewers and improve their primary, secondary, and tertiary prevention capabilities in daily life. At the national level, the release of health videos by elderly doctors through TikTok not only supplements Internet medical resources but also helps to revitalize China’s elderly power resources as advocated in the proposals for the 14th Five-Year Plan for National Economic and Social Development and the Long-Range Objectives Through the Year 2035 [15]. Therefore, it is of considerable practical significance to understand the driving mechanism of TikTok account influence, especially with respect to the accounts of elderly doctors.

Effective communication is an important functional goal of any health system, and social media strengthens the connection between health institutions and individuals and has the potential to strengthen public health communication [16]. In the context of public health initiatives, social media can be used to educate people about health issues, facilitate behavioral changes [17], and form community partnerships for health-related actions [18]. With the constant development of Internet technologies, common forms of computer-mediated communication have extended beyond text, images, and audio to include videos and live streaming [19]. Formats with high media richness make it easier to transfer and understand information [20], thereby increasing trust and interactions in the online environment [21]. Robert and Dennis [22] used distraction conflict theory to study the paradox of media richness; that is, high media richness enhances people’s motivation to process information, in addition to weakening their ability to process information, thus affecting the transmission efficiency of information.

Therefore, it is unclear whether platforms with high media richness, such as TikTok, have higher information transmission efficiency than traditional social media platforms with low media richness, i.e., mainly comprising text and pictures. Thus, it is necessary to study TikTok account influence in social networks, that is, the extent to which an account influences information transmission on the network and affects the views, emotions, and behaviors of others [23,24,25]. Research on the antecedents of account influence mainly consider social and technological characteristics. However, most studies on this topic have examined only a single aspect of the social or technological system through traditional symmetrical variation-oriented methods, and comprehensive studies that include both are lacking. 

In this study, we aim to address the following core research question: how do the TikTok accounts of elderly doctors who publish health-related popular science videos achieve strong influence (or avoid weak influence)? We applied an asymmetric configurational method (i.e., fuzzy-set qualitative comparative analysis, fsQCA) to investigate various configurational solutions of microvideo social media platforms, which are complex interactive systems with technical and social characteristics [26,27,28,29]. 

## 2. Literature Review and Theoretical Framework

### 2.1. Antecedents of Social Media Account Influence

In the existing literature, the antecedents of account influence mainly include social and technological factors. With respect to social aspects, Chai and Kim [30] discussed the knowledge contribution behavior of users on social networking sites. They found that the structural technical assurance of the Internet system (e.g., encryption) did not have a significant impact on people’s knowledge contribution behavior, whereas social ties and the ethical culture of social systems significantly positively affected the knowledge contribution behavior of users. Other studies have shown that the source credibility of social media content positively affects consumers’ consumption and contribution behaviors; the latter involves both peer-to-content, such as commenting on a post, and peer-to-peer interactions, such as forwarding a post to others [31].

With respect to technical aspects, several studies have explored how video format and type affect social media user engagement. Li et al. [8] explored how the format and type of COVID-19-related videos on TikTok affect user engagement. They found that videos with hashtags and subtitles were shared more often and were more likely to spread than those without these features; in terms of video types, dance videos were shared more than slide shows, documentaries, animated infographics, oral speeches, or news. Chung [32] examined the role of Twitter in a smoking cessation campaign and found that the use of conversational tools, such as mentions, reposts, and hashtags in campaign-related tweets made it easier for Twitter users to find relevant information and promoted interaction between Twitter users. Wang et al. [33] studied social user engagement on Facebook and found that the length of a post was positively correlated with the number of shares. Some studies of the antecedents of influence have examined the effects of the richness of media characteristics. Cao et al. [31] studied consumer engagement behavior in social media and found that media richness positively promoted user engagement behavior, including consumption, contribution, and creative behavior. Chen et al. [34] systematically investigated how the Chinese central government used social media (‘Healthy China’, the official Sina Weibo of the National Health Commission of China) to promote citizens’ participation in public health efforts during the COVID-19 crisis. They specifically investigated the role of technical features of social media. They found that high media richness and the provision of a dialogue loop function significantly enhanced citizens’ engagement behavior.

Studies of the antecedents of account influence have examined both social and technical characteristics. However, most studies have examined a single aspect of a social or technical system, and only a small number of integrated analyses have used traditional symmetrical variation-oriented methods. Although the short video social media community is a complex interactive system that includes multiple formats of information technology and a wide variety of social characteristics [26,27,28,29], few studies have used an asymmetric configuration perspective to examine the complex causality in these scenarios.

### 2.2. Sociotechnical Systems Theory

The sociotechnical perspective can be traced back to research conducted by the Tavistock Institute for the British coal industry after World War II [35]. It emerged as a new way of thinking, challenging the dominant view that technology drives social transformation [36]. Cherns [37] outlined a set of principles for sociotechnical design, involving the interrelationships between social and technical elements in an organization. The sociotechnical systems perspective regards any organization as a working system with two interrelated subsystems: a technical system and a social system [38,39,40,41]. The social system mainly consists of the relationships among people and their attributes, such as their skills, attitudes, or values, whereas the technical system consists of the tasks, processes, and technologies that produce specified outputs [38]. In this approach, neither the technical nor the social system is dominant, and an organization is the outcome of the interactions between the two parts [42]. A successful system is a configuration of technical and social factors [30].

As a theoretical framework, the sociotechnical systems theory has been widely used in the field of communications and information systems. Tapia and Maitland [41] used sociotechnical systems theory to explain and predict the selection and use of technology in humanitarian relief and development organizations. Chai and Kim [30] used this framework to discuss the knowledge contribution behavior of users on social networking sites. Wan et al. [43] used this perspective to study how attachment affects users’ willingness to donate to social media content creators.

Building on these studies, in the present research, we adopted the sociotechnical systems approach and used the fsQCA method to examine the conditional configurations of social and technical elements that affect account influence. We aimed to identify the conditions that give the TikTok accounts of some elderly doctors the power to influence people’s thoughts, feelings, and actions based on the TikTok system interface. The research framework is shown in Figure 1.

## 3. Research Methods

### 3.1. Sample

For the configurational analysis, we used a sample of 63 accounts of elderly doctors (older than 60 years, with a maximum age of 92 years) on the TikTok platform as of February 2021. The doctors were mainly from eastern cities, such as Beijing, Shanghai, Guangzhou, Jinan, and Nanjing, with a smaller number of accounts from Zhengzhou, Wuhan, and Changsha in central China and Xi’an in western China. The sample included doctors from public and private hospitals working in various medical departments, including internal medicine, gynecology, urology, orthopedics, ophthalmology, endocrinology, pediatrics, and dermatology, among others (Table 1).

### 3.2. Fuzzy-Set QCA Procedures, Methods, and Data Preparation

We used fsQCA for our analyses and sufficient causality tests. FsQCA is well-suited for asymmetric and complex configurational problems [44]. This method is derived from complexity theory and is based on the principles of conjunction, equality, and causal asymmetry [45]. The conjunction criterion means that the conditions operate interdependently rather than discretely or through simple two- or three-way interactions between those conditions. Equality implies that multiple configurations may lead to the same outcome, and causal asymmetry means that a condition that leads to a particular outcome in one configuration may be completely different in another configuration that leads to the same result [44,46].

We reported our results using the following notations: ● and ⊗ represent the presence and absence, respectively, of core conditions; 🞄 and ^⊗^ represent the presence and absence, respectively, of peripheral conditions; and a blank space indicates that the presence or absence of the focal condition did not affect the outcomes [47]. Core conditions are those that were present in both the intermediate and parsimonious solutions (or necessary conditions). Peripheral conditions are those that were present in the intermediate solutions but not in the parsimonious solutions [47].

### 3.3. Measures and Calibrations for Set Membership

Before conducting the configurational analysis, we calibrated the variable for each condition. Calibration is a process of assigning degrees of membership in sets [48]. This study involved two kinds of sets: crisp sets (0 or 1; non-membership = 0; full membership = 1) and fuzzy sets (between 0 and 1). According to Ragin [44], calibration for fuzzy sets usually involves the use of three thresholds to convert variables into membership sets: full membership (set membership = 1), the crossover point (set membership = 0.5), and full non-membership (set membership = 0). There are two main methods of calibration for fuzzy-sets: direct calibration and indirect calibration. The direct method relies on the setting of the values associated with the three qualitative anchors (the threshold for full membership, full non-membership, and the crossover point), and the indirect method relies on the researcher broadly grouping cases according to their membership in the target set. 

In this study, we directly assigned 0 or 1 to calibrate the crisp-set membership and used both the direct and indirect methods to calibrate the fuzzy-set membership.

#### 3.3.1. Outcome Variable and Calibration

We measured account influence using an index provided by the Douchacha platform, i.e., an indicator variable that reflected the total value and influence of an account, including the number of likes, the number of shares, and other data, up to a given point in time. Table 2 shows the thresholds for the variables in this study. With reference to the work of other scholars [44,47,48], we used the direct calibration method to compute percentiles, with the upper 25th percentile serving as the threshold for full membership (1013), the 50th percentile representing the cross-over point (940), and the lower 25th percentile serving as the threshold for full non-membership (860.5).

#### 3.3.2. Conditional Variables and Calibrations

Several of the variables were crisp sets, such as oncology-related (OR, meaning the profile on the home page of the doctor’s account mentioned oncology or cancer), public attribute (PA, meaning that the medical institution listed on the home page of the doctor’s account was a public hospital), comment interaction (CI, meaning that the doctor answered questions from fans in the comments section of the account), and video collection (VC, meaning that the home page of the account had a collection of videos). Using the direct assignment method, we set full membership (that is, the condition existed) as 1 and full non-membership (that is, the condition did not exist) as 0.

A three-value set is a basic type of fuzzy set. As shown in Table 2, instead of using only two scores (0 and 1), variables such as telepresence (TP) were assigned a third value of 0.5 to identify objects that were neither entirely in nor entirely outside of the set. We used the indirect calibration method to calibrate TP according to the actual situation. TP refers to the feeling of immersion created by the medium [49,50,51]. Angles and distances are important determinants of TP [52]. Therefore, TikTok videos that use a variety of shooting angles and lens distances (e.g., a close-up shot of an interaction between a doctor and a patient) were assigned a TP value of 1, and videos that paid no attention to shooting angles and lens distances were assigned a value of 0 (e.g., a video in which the doctor only talks to the camera). Videos with some variation in shooting angles and lens distances were assigned a value of 0.5 (e.g., occasional interaction between a doctor and patient). Furthermore, with reference to previous studies [53], we changed 0.5 to 0.499 because if a condition is assigned a value of 0.5, the fsQCA software will automatically deletes it during the analysis process.

## 4. Results

### 4.1. Necessary Conditions Analysis

Following established QCA practices, we first conducted a fuzzy-set analysis of the necessary conditions (see Table 3) using a consistency benchmark of 0.90. We found no pathways in which a single factor of OR, PA, CI, TP, or VC showed a strong influence. 

### 4.2. Sufficiency Analysis of Account Influence

Following established QCA procedures, we conducted a sufficiency analysis using a frequency benchmark (≥1), a raw consistency benchmark (≥0.8), and a proportional reduction in inconsistency (PRI; ≥0.70) [54]. Here, we report two sets of results: the configurations associated with the presence of strong account influence and the configurations associated with the absence of strong account influence (see Table 4). We identified various configurations of sociotechnical elements that can create influential accounts; a scenario with multiple solutions can be said to have achieved equifinality, allowing us to move beyond the either/or paradigm used in previous literature to explain account influence.

#### 4.2.1. Configurations for Strong Account Influence

The two solutions for strong account influence are presented in Table 4. In the first solution (S1), PA was a peripheral condition, the presence of VC and the absence of CI or TP were core conditions, and OR had no effect on account influence. In other words, despite the lack of participation in comment interaction and the poor sense of telepresence, strong account influence was achieved if the account belonged to a public hospital and a video collection setup existed. In the second solution (S2), OR was a core condition, TP and VC were core conditions, and PA was a peripheral condition; however, CI was irrelevant to strong account influence. In other words, despite a poor sense of telepresence, strong account influence was still possible as long as the account belonged to an oncology-related department in a public hospital that also had a video collection. This suggests that CI and OR are, to some extent, exchangeable across these two solutions.

The results in Table 4 indicate an overall solution coverage of 0.093, which suggests that the two solutions explain about 10 per cent of highly influential accounts. The overall solution consistency indicates that the level of sufficiency, at 0.973, is substantially higher than the generally accepted threshold of 0.75, as suggested by Ragin [44]. Raw coverage describes the proportion of the results explained by a specific solution, and unique coverage describes the proportion of the result exclusively specified by a solution. As shown in Table 4, the raw coverage for a single solution ranges from 0.045 to 0.064, and the unique coverage for a single solution ranges from 0.029 to 0.048, with a corresponding consistency ranging from 0.947 to 1. Therefore, all of the solutions exceed the required threshold.

#### 4.2.2. Configurations for Absence of Strong Account Influence

In solution NS1, PA, TP, and VC were core conditions, and they were absent; CI was a peripheral condition and was present; and OR was an unimportant condition. In NS2, PA was present and peripheral, OR was present and core, TP and VC were absent and core, and CI was an unimportant condition. In NS3, the peripheral condition PA was present, the peripheral condition OR was absent, the core condition of CI was absent, and the core conditions of TP and VC were present. The results also show the configurational effects of different combinations of antecedents on the negation of SAI.

The results indicate an overall solution coverage of 0.155, which suggests that a certain proportion of high account influence could be explained by the three aforementioned solutions. The overall solution consistency value of 0.980 was significantly above the threshold of 0.75. The raw and unique coverage for a single solution ranged from 0.029 to 0.079, with the corresponding consistency ranging from 0.910 to 1. Therefore, all of the solutions met the required threshold.

## 5. Discussion

### 5.1. Theoretical Implications

Our study makes several contributions to the literature. First, applying the sociotechnical system approach to health science videos on social media, we developed an integrated analysis framework to explain the mechanisms that drive account influence. Secondly, we identified a number of configurations of sociotechnical elements that can create influential accounts; a scenario with multiple solutions can be said to have achieved equifinality, allowing us to move beyond the either/or paradigm to explain account influence. Thirdly, we extended insights from previous studies on social media influencers by considering causal asymmetry [44], revealing new asymmetric pathways between strong and not-strong influence. Fourthly, in this study, we used a sociotechnical system framework and a ‘configuration perspective’ to identify the synergistic effects of varying combinations of conditions. This further extends the application of the sociotechnical system framework to encompass ‘causal complexity’.

### 5.2. Practical Implications

The results of the present study have practical implications for people trying to increase their reputation and the influence of their short video social media accounts. In particular, the reported results provide insights to help elderly doctors actively develop the human resources necessary to reach out to the aged in China. In general, the study results demonstrate which configurations of social and technical factors should be strengthened to increase the influence of an account. Furthermore, aspects of the social system, especially account attributes, can be improved; for example, professional authentication and platform audits should be strengthened to enhance the credibility of accounts. In terms of the technical system, systematic classification of account content could be implemented to improve the legibility of accounts.

We used fsQCA to analyze the TikTok accounts of 63 elderly Chinese doctors and showed that relationships between different sociotechnical configurations and account influence are causally asymmetric. We found that multiple configurations of social and technical conditions formed two equifinal paths to strong account influence and that three pathways negated strong account influence. This confirms the causal complexity of the relationship among social systems, technical systems, and account influence in our research context. Health science communication accounts on microvideo social media platforms are best-positioned to succeed when they avoid a one-size-fits-all strategy. To succeed, these accounts should rely on varying combinations of elements of the social and technical systems that form microvideo social media platform environments.

### 5.3. Limitations and Future Research

We must acknowledge some limitations of the current study, which should be addressed in future research. First, the sample size was relatively small, and the samples all came from one single platform, i.e., TikTok. The number of possible configurations grows exponentially (i.e., 2^N^) with the number of conditions included, so the more causal conditions there are, the more the problem of “limited diversity” results. In addition to TikTok, a number of other microvideo social media platforms host health science communication accounts, such as KuaiShou, Tencent WeiShi, and WeChat Channels. Therefore, future studies should involve a larger heterogeneous sample that draws from multiple platforms. 

Secondly, the consistency and coverage scores of fsQCA indicated the percentage of results explained by each solution; however, the percentage not explained by the solutions indicates the need to analyze the impact of conditions not included in this study. Therefore, in addition to increasing the heterogeneous sample, future studies can further expand the model design by integrating other theoretical perspectives and including other equally important condition variables. 

Thirdly, in this study, we used cross-sectional data, which did not reflect dynamic attributes. We did not integrate temporal effects into our model, including dynamic characteristics, such as user increment and information propagation of microvideo social media platforms [25,55,56]. Future research should involve the collection of more longitudinal data in order to analyze time effects under the fsQCA framework.

## 6. Conclusions

In this paper, we used fsQCA to detect the effects of different configurations of three social system characteristics (i.e., oncology-related attribute, public attribute, and comment interaction) and two technical system characteristics (i.e., telepresence and video collection) on TikTok accounts. Our results reveal two pathways associated with distinct sociotechnical configurations to strong account influence and three pathways associated with distinct sociotechnical configurations to not-strong account influence. 

For the two configurational solutions of strong account influence, first, regardless of whether the doctor belongs to an oncology-related department, even if the interaction atmosphere and telepresence are not good, as long as the doctor is from a public hospital and the setting of video classification and collection is added to the TikTok page, strong account influence can be achieved. Secondly, regardless of whether we pay interaction with users’ comments is considered, even if the feeling of presence is not good, as long as the home page has the setting of video classification collection, the hospital is a public hospital, and the doctor belongs to an oncology-related department, strong account influence can be achieved.

For the three configurational solutions for absence of strong account influence, first, regardless of whether the doctor belongs to an oncology-related department, even if the interactive atmosphere is considered, strong account influence will not be achieved as long as the doctor is not from a public hospital, has a poor sense of presence, and has no setting of video classification and collection. Secondly, regardless of whether interaction with users’ comments is considered, even if the doctor belongs to an oncology-related department and comes from a public hospital, as long as telepresence is not good and there is no setting of video classification and collection on the TikTok page, the influence of the account will not be strong. Thirdly, even if there is a good sense of telepresence and the setting of video classification and collection and the doctor is from a public hospital, as long as the doctor does not belong to an oncology-related department and does interact with users’ comments, the account influence will not be strong.

Our results confirm that a single antecedent condition cannot, on its own, produce an outcome. Multiple interrelated conditions are required to produce an influential account. These results offer a more holistic picture of how health science communication accounts operate, reconciling the scattered results in the literature. 

## Figures and Tables

**Figure 1 ijerph-19-13815-f001:**
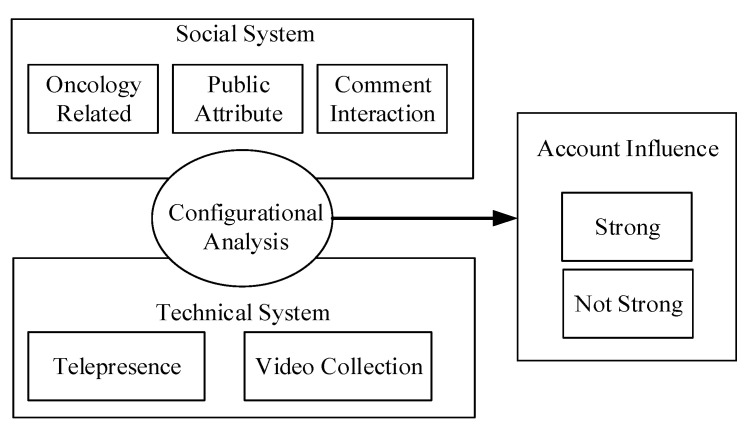
Research mode.

**Table 1 ijerph-19-13815-t001:** Sample summary.

Medical Departments (Number of Doctors)	Cities (Number of Doctors)	Age (Frequency)
Gynecology and Obstetrics (12)	Beijing (44)	60–65 (24)
Traditional Chinese Medicine (10)	Shanghai (5)	66–70 (8)
Andrology and Urology (6)	Zhengzhou (4)	71–75 (13)
Oncology (6)	Guangzhou (3)	76–80 (8)
Internal Medicine (4)	Changsha (2)	81–85 (6)
Pediatrics (4)	Wuhan (2)	86–90 (3)
Nephrology (4)	Nanjing (1)	91–95 (1)
Endocrinology (2)	Jinan (1)	
Cardiovascular (2)	Xi’an (1)	
Health Care (2)		
Orthopedics and Sports Medicine (2)		
Other (Ophthalmology/Dermatology/Breast Surgery/Otorhinolaryngologic/etc.) (9)		

**Table 2 ijerph-19-13815-t002:** Variables and calibration types.

Variable	Thresholds		
	Full Non-Membership	Cross-Over Point	Full Membership
OR	0	–	1
PA	0	–	1
CI	0	–	1
TP	0	0.499	1
VC	0	–	1
Account influence	860.5	940	1013

Note: ‘–’ represents blank. We changed 0.5 to 0.499 because the fsQCA software automatically deletes observations with a value of 0.5.

**Table 3 ijerph-19-13815-t003:** Necessary conditions analysis.

Set of Conditions	Strong Account Influence		~Strong Account Influence	
	Consistency	Coverage	Consistency	Coverage
OR	0.216	0.451	0.260	0.549
~OR	0.784	0.513	0.740	0.487
PA	0.837	0.505	0.814	0.495
~PA	0.163	0.465	0.186	0.535
CI	0.881	0.532	0.771	0.468
~CI	0.119	0.341	0.229	0.659
TP	0.527	0.518	0.582	0.575
~TP	0.567	0.574	0.512	0.521
VC	0.575	0.623	0.346	0.377
~VC	0.425	0.392	0.654	0.608

Note: ~ means the absence of; for example, ~OR = absence of OR.

**Table 4 ijerph-19-13815-t004:** Configurations for achieving strong account influence or absent influence.

Causal Condition	SAI		Absence of SAI		
	S1	S2	NS1	NS2	NS3
Social System					
OR		●		●	^⊗^
PA	🞄	🞄	⊗	🞄	🞄
CI	⊗		🞄		⊗
Technical System					
TP	⊗	⊗	⊗	⊗	●
VC	●	●	⊗	⊗	●
Consistency	0.9467	1	0.9960	1	0.91
Raw coverage	0.0453	0.0638	0.0789	0.0476	0.0288
Unique coverage	0.0293	0.0478	0.0789	0.0476	0.0288
Overall solution consistency	0.9734		0.9800		
Overall solution coverage	0.0931		0.1552		

Note: SAI = strong account influence; S1 and S2 are abbreviations of SAI; NS1, NS2, and NS3 are abbreviations of negation of SAI; ● = core causal condition present; ⊗ = core causal condition absent; 🞄 = peripheral condition present; ^⊗^ = peripheral condition absent.

## Data Availability

The raw data supporting the conclusions of this article will be made available by the authors, without undue reservation.

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
