# Peer review of "What Drives the Influence of Health Science Communication Accounts on TikTok? A Fuzzy-Set Qualitative Comparative Analysis"

_ijerph, 2022, doi:10.3390/ijerph192113815_

Round 1
Reviewer 1 Report
This paper provide an interesting topic - usage of social media in healthcare - especially TikTalk
Suggestions:
underline the aim of study
clarify the choice and the reason why the focus of the research is on Tik Talk and not on another social network
whether there is a procedure related to the protection of patient data when using this social network
the limitations of the study are explained
it is not clear whether this social network is suggested as an official channel of communication in the function of information of importance for public health
what are the risks and dangers of using social networks
Reviewer 2 Report
The paper presented is of high quality and methodologically well thought out. Nevertheless, I think that the introduction could be made shorter and a section on the literature review should be created and developed. On the other hand, it is not enough to say that the results contribute to the literature review, it is necessary to conform these data to what exists in the literature review.
On page 4, there is a figure on research mode, which I think would be better in a chapter on work methodology and not at the end of the introduction.
Finally, the questions I was going to ask about the research design of this paper have already been presented by the authors in the chapter on limitations and future research. I think this chapter should come after the conclusions and not before.
The conclusions should also be improved, because what is there is too similar to the abstract text.
